# The Angiotensin II Receptor Blocker Losartan Sensitizes Human Liver Cancer Cells to Lenvatinib-Mediated Cytostatic and Angiostatic Effects

**DOI:** 10.3390/cells10030575

**Published:** 2021-03-05

**Authors:** Hirotetsu Takagi, Kosuke Kaji, Norihisa Nishimura, Koji Ishida, Hiroyuki Ogawa, Hiroaki Takaya, Hideto Kawaratani, Kei Moriya, Tadashi Namisaki, Takemi Akahane, Akira Mitoro, Hitoshi Yoshiji

**Affiliations:** Department of Gastroenterology, Nara Medical University, 840 Shijo-cho, Kashihara, Nara 634-8521, Japan; htakagi@naramed-u.ac.jp (H.T.); nishimuran@naramed-u.ac.jp (N.N.); ishidak@naramed-u.ac.jp (K.I.); ogawah@naramed-u.ac.jp (H.O.); htky@naramed-u.ac.jp (H.T.); kawara@naramed-u.ac.jp (H.K.); moriyak@naramed-u.ac.jp (K.M.); tadashin@naramed-u.ac.jp (T.N.); stakemi@naramed-u.ac.jp (T.A.); mitoroak@naramed-u.ac.jp (A.M.); yoshijih@naramed-u.ac.jp (H.Y.)

**Keywords:** HCC, VEGF, angiotensin-II, angiogenesis, lenvatinib

## Abstract

Molecular targeted therapy with lenvatinib is commonly offered to advanced hepatocellular carcinoma (HCC) patients, although it is often interrupted by adverse effects which require a reduction in the initial dose. Thus, an alternative lenvatinib-based therapy to compensate for dose reduction is anticipated. This study aimed to assess the effect of combination of low-dose of lenvatinib and the angiotensin-II (AT-II) receptor blocker losartan on human HCC cell growth. In vitro studies found that losartan suppressed the proliferation by inducing G1 arrest and caused apoptosis as indicated by the cleavage of caspase-3 in AT-II-stimulated HCC cell lines (Huh-7, HLE, and JHH-6). Losartan attenuated the AT-II-stimulated production of vascular endothelial growth factor-A (VEGF-A) and interleukin-8 and suppressed lenvatinib-mediated autocrine VEGF-A production in HCC cells. Moreover, it directly inhibited VEGF-mediated endothelial cell growth. Notably, the combination of lenvatinib and losartan augmented the cytostatic and angiostatic effects of the former at a low-dose, reaching those achieved with a conventional dose. Correspondingly, a HCC tumor xenograft assay showed that the oral administration of losartan combined with lenvatinib reduced the subcutaneous tumor burden and intratumor vascularization in BALB/c nude mice. These findings support that this regimen could be a viable option for patients intolerant to standard lenvatinib dosage.

## 1. Introduction

Hepatocellular carcinoma (HCC) accounts for more than 90% of primary liver cancer, which is the fourth leading cause of cancer-related mortalities worldwide [1,2,3]. The overall ratio of mortality to incidence in HCC is 0.95 and this poor prognosis has shown no satisfactory improvements, regardless of disease etiology [4]. Clinically, HCC is asymptomatic at an early stage, which leads to diagnostic delays, with patients diagnosed at the advanced stage often being ineligible for curative surgery, and a limited availability and efficacy of therapeutic options for advanced HCC patients. Moreover, the vast majority of HCC is developed in cirrhotic liver with a loss of hepatic function, worsening prognosis [5,6,7].

The latest clinical guidelines by the American Society of Clinical Oncology recommend offering atezolizumab-bevacizumab (Atezo + BV) as first-line treatment for patients with advanced HCC, although the tyrosine kinase inhibitors (TKI), sorafenib or lenvatinib, may be offered as first-line treatment in the presence of contraindications to Atezo + BV [8]. Among these therapeutic agents, lenvatinib is an oral, small-molecule TKI that targets vascular endothelial growth factor (VEGF) receptor (VEGFR) 1-3, fibroblast growth factor (FGF) receptor (FGFR) 1–4, platelet-derived growth factor receptor (PDGFR) α/β, KIT, and RET [9]. It exerted a potent anticancer effect in the Multicenter, Randomized, Openlabel, Phase 3 Ttial to Compare the Efficacy and Safety of Lenvatinib Versus Sorafenib in First-Line Treatment of Subjects With Unresectable Hepatocellular Carcinoma (REFLECT ) trial, showing that the objective response rate, and disease control rate were 24.6 and 75.5%, respectively, better outcomes than those achieved with sorafenib [10]. Currently, although lenvatinib is most frequently used for advanced unresectable HCC in Japanese clinical practice, it is hard for many patients to be continuously medicated with lenvatinib long periods due to tumor progression and adverse events (AE), including fatigue, appetite loss, and hypertension [11]. Therefore, newly developed molecular targeting agents with acceptable AEs for long-term usage are desirable.

Angiotensin II (AT-II) is a peptide hormone known as a vasoconstrictor with central roles in hypertension, heart failure, and chronic renal injury [12,13,14]. It is typically generated by the removal of two residues from angiotensin-I by the angiotensin-converting enzyme (ACE). AT-II is a ligand for two receptors, AT-II type 1 receptor (AT1R) and AT2R, each evoking distinct signaling pathways and physiological responses [15]. ACE inhibitor (ACE-I) and AT1R blocker (ARB) blocking the biological activities of AT-II are commonly used as anti-hypertensive drugs without serious AEs in clinical practice. Several lines of evidence have suggested that these agents could show potent anti-angiogenic properties and suppress the tumor growth of several malignancies, including HCC, at clinically equivalent doses [16,17,18]. We have demonstrated that pharmacological blockade of AT-II and AT1R signaling by these agents significantly suppressed tumor growth and angiogenesis with reduced intratumor VEGF production in a mouse HCC allograft model [19]. A recent report also has shown that ARB efficiently attenuated AT-II-induced HCC cell proliferation by inhibiting the AT1R/Raf/extracellular signal-regulated kinase (ERK)1/2 signaling pathway [20]. Moreover, a recent clinical cohort study has suggested that patients treated with sorafenib plus ACE-I/ARB had an increased median overall survival compared to those treated with sorafenib alone [21]. However, the efficacy of ARB to modulate the antitumor activity of lenvatinib against HCC growth and the mechanism underlying ARB-mediated inhibition of cell proliferation and intratumor angiogenesis are unknown.

Here, we examined the effect of combination of lenvatinib and ARB on the tumor growth of human HCC cells in a mouse xenograft model. Especially, we investigated whether the antitumor effect of lenvatinib at a low dose plus ARB was comparable to that of conventional lenvatinib dose, aiming to explore if the addition of ARB could compensate for a lenvatinib dose reduction in the clinical practice.

## 2. Materials and Methods

### 2.1. Compounds and Cell Culture

Lenvatinib was obtained from ChemScene (Monmouth Junction, NJ, USA), and losartan potassium was supplied by Merck and Co., Inc. (Kenilworth, NJ, USA). Human AT-II acetate salt was purchased from Wako Pure Chemical Corporation (Osaka, Japan). Human liver cancer cell lines (Huh-7, HLE, and JHH-6) and human umbilical vascular endothelial cells (HUVEC) were procured from the Japanese Collection of Research Bioresources Cell Bank (Osaka, Japan). The cells were grown and maintained at 37 °C in Dulbecco’s modified Eagle’s medium (DMEM) (Nacalai tesque, Kyoto, Japan) containing 10% Fetal bovine serum (FBS) (Thermo Fisher Scientific Inc., Waltham, MA, USA), 2 mM glutamine, 100 IU/mL penicillin, and 100 μg/mL streptomycin as the culture medium, plated on 100 mm cell culture dishes, and incubated at 37 °C in a 5% CO_2_ air environment. Mycoplasma testing was performed using the MycoProbe^®^ Mycoplasma Detection Kit (R&D Systems, Inc., Minneapolis, MN, USA) according to the manufacturer’s protocol.

### 2.2. Cell Proliferation Assay

Huh-7, HLE, and JHH-6 cells were seeded in 96-well plates with DMEM and 1% FBS for 24 h. Then, cells were treated as follows: (1) different concentrations of lenvatinib (0–10^−5^ M) for different durations (0–6 days), (2) pre-treatment with different concentrations of AT-II (0–1 μM) for 12 h and subsequent treatment with losartan (1 μM) for 12 h, and (3) pre-treatment with AT-II (1 μM) for 12 h and subsequent treatment with losartan (1 μM) and lenvatinib (1 or 3 μM) for 12 h. HUVEC cells were also seeded in the same conditions as liver cancer cells and were then stimulated with VEGF-A (10 ng/mL) and concomitantly treated with lenvatinib (1 μM) and vehicle (DMSO)/ amlodipine (Wako Pure Chemical Corporation, Osaka, Japan: 10 μM)/ propranolol (LKT Laboratories, St Paul, MN, USA: 50 μM)/ losartan (1 μM) or with lenvatinib (3 μM) alone for 24 h. The Bromodeoxyuridine (BrdU) Cell Proliferation ELISA (Cosmo Bio, Tokyo, Japan) was used to evaluate cell proliferation according to the manufacturer′s protocol.

### 2.3. Measurement of Cleaved Caspase-3

To assess in vitro cell apoptosis, cleaved caspase-3 concentration in cell extracts from human liver cancer cells was measured using the Human Cleaved Caspase-3 (Asp175) ELISA (AbCam, Cambridge, UK) according to the manufacturer′s instructions. A total of 1 × 10^6^ cells were pre-treated with AT-II (1 μM) for 12 h and then treated with lenvatinib (1 or 3 μM) and/or losartan (1 μM) for 12 h following overnight starvation.

### 2.4. Measurement of VEGF-A, IL-8, and FGF2 Levels

Vascular endothelial growth factor (VEGF)-A, Interleukin (IL)-8, and Fibroblast growth factor (FGF) 2 concentrations in the cultured media from human liver cancer cells were measured using the RayBio Human VEGF-A ELISA Kit (RayBiotech, Inc., Peachtree Corners, GA, USA), the Human IL-8 ELISA Kit (Abcam), and the Human FGF basic ELISA Kit (Abcam) according to the manufacturer′s instructions. A total of 1 × 10^6^ human liver cancer cells were treated with different concentrations of AT-II (10^−2^–1 μM) for 12 h, pre-treated with AT-II (1 μM) for 12 h and then treated with losartan (1 μM) for 12 h, or losartan (1 µM) and/or lenvatinib (1 or 3 μM) for 24 h following overnight starvation.

### 2.5. In Vitro Endothelial Tubular Formation

In vitro endothelial tubular formation was defined as the formation of capillary-like structures in co-cultures of HUVECs with normal human dermal fibroblasts, as described previously [18,22]. HUVECs were seeded to each well at 2.5 × 10^4^ and incubated at 37 °C for 20 h in a 5% CO_2_ atmosphere and were then stimulated with VEGF-A (10 ng/mL) and concomitantly treated with lenvatinib (1 μM) and vehicle (DMSO)/ amlodipine (10 μM)/ propranolol (50 μM)/ losartan (1 μM) or with lenvatinib (3 μM) alone for 24 h. A semiquantitative analysis of tubule formation was performed using the ImageJ software ver1.52 (http://imagej.nih.gov/ij/).

### 2.6. RNA Extraction and Reverse Transcription-quantitative Polymerase Chain Reaction (RT-qPCR)

Total RNA was isolated from liver tissues and 10^6^ cultured Huh-7, HLE, and JHH-6 cells using the RNeasy Mini Kit (Qiagen, Hilden, Germany). The resulting RNA concentrations were determined using a NanoDropTM 2000c Spectrophotometer (Thermo Fisher Scientific Inc.). The High-Capacity RNA-to-cDNA kit (Applied Biosystems, Foster City, CA, USA) was used for reverse transcription to generate cDNA. Quantitative RT-PCR (qRT-PCR) was performed with the primer pairs described in Appendix A using a SYBR^TM^ Green PCR Master Mix (Applied Biosystems) and an Applied Biosystems StepOnePlus™ Real-Time PCR^®^ system (Applied Biosystems). Relative expression levels were normalized to glyceraldehyde 3-phosphate dehydrogenase (*GAPDH*)expression and estimated using the 2^-ΔΔCT^ method and presented as fold changes relative to controls.

### 2.7. Human Liver Cancer-Derived Xenograft

The six-week-old male athymic nude mice (BALB/c Slc-nu/nu) were purchased from Japan SLC, Inc. (Shizuoka, Japan). Mice rearing and inoculation of Huh-7 human liver cancer cells were performed, as previously described [18]. Then, 10 days after inoculation, the interventional mice were distributed into the following six treatment groups (*n* = 10 in each group): vehicle, Los (losartan, 30 mg/kg), Lv-LD (lenvatinib, 3 mg/kg), Lv-HD (lenvatinib, 10 mg/kg), Lv-LD (3 mg/kg) + Los (30 mg/kg), and Lv-HD (10 mg/kg) + Los (30 mg/kg), and were administered treatment through daily oral gavage as monitoring their tumor volumes [18]. The dosages of losartan and lenvatinib for mice were defined according to previous reports [18,23,24]. All mice were sacrificed 21 days after administration and their subcutaneous tumors were then collected. Serum biological markers were measured using routine laboratory methods. All animal procedures complied with the recommendations of the Guide for Care and Use of Laboratory Animals (National Research Council of Japan), and the study was approved by the ethics committee of Nara Medical University, Kashihara, Japan (Authorization No. 12767).

### 2.8. Histological and Immunohistochemical Analyses

Resected tumor tissues were fixed overnight at 4 °C in 10% formalin and embedded in paraffin. Subsequently, 5 µm paraffin sections were routinely stained with hematoxylin and eosin. Immunohistochemical analyses were also performed using paraffin-embedded tumor sections as described [18]. As the primary antibodies, rabbit anti-Ki67 (Abcam, Cambridge, England; 1:100 dilution) and rabbit anti-CD34 (Abcam; 1:2500 dilution) were used, with staining performed according to the suppliers’ recommendations. 

TdT-mediated dUTP Nick End Labeling (TUNEL)-positive cells in subcutaneous tumor sections were detected by using an In-Situ Cell Death Detection Kit (Sigma–Aldrich, St. Louis, MO, USA), as recommended for tissue sections by the supplier. Ki67-positive cells and TUNEL-positive cells were counted in high-power fields (HPF) at 400-fold magnification. CD34-positive areas were quantified in HPFs at 400-fold magnification using the ImageJ software. All quantitative analyses were performed for five fields per each section.

### 2.9. Measurement of AT1R Protein Levels

After equalizing the protein concentration from frozen subcutaneous tumor samples to 5 mg/mL, AT1R protein levels were measured using the Human Angiotensin II Receptor 1 ELISA Kit (MyBioSource, Inc., San Diego, CA, USA) according to the manufacturer’s instructions. Quantitative values were relatively indicated as fold change to the value of total protein from human adult liver tissue (BioChain Institute Inc., Newark, CA, USA).

### 2.10. Statistical Analyses

Statistical analyses were performed using Prism, version 9 (GraphPad Software, La Jolla, CA, USA). Data are expressed as the mean ± standard deviation. Statistical variance between each experimental group was analyzed using an analysis of variance test. Bartlett′s test was used to determine the homogeneity of variances. All tests were two-tailed and *p*-values < 0.05 were considered statistically significant.

## 3. Results

### 3.1. Effects of Lenvatinib and Losartan on in vitro Human Liver Cancer Cell Growth.

Initially, we assessed the effects of combination of lenvatinib and losartan on in vitro human liver cancer cell growth. To optimize the concentrations of lenvatinib used for in vitro studies, we confirmed the anticancer effects of lenvatinib at different doses in three liver cancer cell lines, Huh-7, HLE, and JHH-6. As shown in Figure 1A, lenvatinib efficiently suppressed the growth of these liver cancer cells in a dose-dependent manner at 0.1–10 μM. We did not observe a significant suppression of human liver cancer cell proliferation through losartan treatment under normal culture conditions (Appendix A). Meanwhile, losartan treatment significantly inhibited AT-II-stimulated cell proliferation in liver cancer cells (Figure 1B). Consistently, the addition of losartan enhanced the lenvatinib-mediated inhibitory effects on AT-II-stimulated cell proliferation of human liver cancer cells. Notably, the anti-proliferative effect of losartan combined with lenvatinib (1 μM) was approximately equivalent to that of lenvatinib (3 μM) (Figure 1C). Regarding the molecular basis of cell cycle arrest, lenvatinib treatment significantly decreased the expression of Cyclin D1 (*CCND1*), Cyclin-dependent kinase-4 (*CDK4*), *CCNE1*, and *CDK2*, which contribute to the regulation of the G_1–_S phase transition. Decreases in these mRNA levels were augmented by combination with losartan (Figure 1D and Appendix A). Regarding apoptosis, treatment with lenvatinib alone did not change the levels of cleaved caspase-3 and the mRNA expressions of apoptosis-related markers in liver cancer cells (Figure 1E,F and Appendix A). On the other hand, the combination of lenvatinib and losartan significantly increased the levels of cleaved caspase-3 in human liver cancer cell culture extracts (Figure 1E). This was evidenced by changes in the mRNA levels of apoptosis-related markers (decreased anti-apoptotic *BCL2* and increased pro-apoptotic Bcl-2-associated X protein (*BAX*) and *BAK* expression). (Figure 1F).

### 3.2. Effects of Lenvatinib and Losartan on Angiogenic Activity in Human Liver Cancer Cells and HUVECs.

To examine the effect of losartan on angiogenic activity in human liver cancer cells, we evaluated VEGF-A, IL-8, and FGF2 production by AT-II stimulation in liver cancer lines. As shown in Figure 2A, AT-II stimulation increased VEGF-A and IL-8 production at a dose of 1 μM in Huh-7, HLE, and JHH-6 cells, while FGF2 levels were not altered by the AT-II stimulus. AT-II-stimulated overproductions of VEGF-A and IL-8 were significantly suppressed by treatment with losartan, while FGF2 levels were unaltered with losartan treatment (Figure 2B–D). Recent reports have suggested that inhibition of TKs, including VEGFR and FGFR, induces the autocrine production of their ligands in cancer cells [25,26]. Thus, we next examined whether treatment with lenvatinib could increase these pro-angiogenic factors in liver cancer cells. As with other TKIs, treatment with lenvatinib significantly increased both VEGF-A and FGF2 production, and we found that losartan reversed the increased production of VEGF-A but not of FGF2 (Figure 2E and Appendix A). These findings support that losartan could exert anti-angiogenic activities in human HCC cells by inhibiting AT-II and AT1R signaling pathway. Next, we assessed the effect of combination of lenvatinib and losartan on the VEGF-induced growth of HUVECs. Based on pharmacological action, treatment with lenvatinib (1 μM) attenuated VEGF-A-induced HUVEC proliferation (Figure 2F). Interestingly, combination with losartan significantly augmented the HUVEC proliferation suppression mediated by lenvatinib (1 μM) (Figure 2F). Moreover, treatment with lenvatinib (1 μM) inhibited the VEGF-induced tubular formation of HUVECs, and this effect was also reinforced by combination with losartan (Figure 2G). Notably, this combination achieved an enhancement of the anti-angiogenic activity of lenvatinib (1 μM) to the same extent of that of lenvatinib (3 μM) (Figure 2F,G). Meanwhile, lenvatinib-mediated effects on HUVEC growth were not modulated by other anti-hypertensive agents, including amlodipine and propranolol (Figure 2F,G).

### 3.3. Losartan Augments the Reduction in Xenograft Tumor Burden Mediated by Lenvatinib in Liver Cancer Cells

We next explored the effects of lenvatinib and losartan on the in vivo growth of xenograft HCC tumors in athymic nude mice, as shown in Figure 3A. In this model, lenvatinib and losartan administration did not impair liver and renal functions (Figure 3B). Huh-7 cell-derived xenograft tumors grew progressively in the control mice, and at a slow rate in mice treated with lenvatinib (3 or 10 mg/kg) or losartan (30 mg/kg) (Figure 3C). The combination of both agents suppressed subcutaneous tumor growth more potently than either single agent (Figure 3C). At the end of the experiments, the mean tumor volumes were significantly smaller in lenvatinib and losartan-treated mice than in those treated by either single agent. The combination of losartan with lower dose of lenvatinib (3 mg/kg) showed a reduction effect comparable to that achieved by a higher dose of lenvatinib (10 mg/kg) (Figure 3D). Correspondingly, the mean tumor weights were significantly decreased by treatment with lenvatinib and losartan in a tendency similar to that of tumor volumes (Figure 3D). We next quantitatively evaluated intratumor cell proliferation and apoptosis by immunohistochemistry. As shown in Figure 4A,B, Ki67-positive cell proliferation was potently reduced by lenvatinib (3 or 10 mg/kg) treatment. Treatment with losartan alone attenuated intratumor cell proliferation in an imperceptible manner. Treatment in combination with both doses of lenvatinib enhanced the attenuation of tumor cell proliferation. TUNEL-positive cell apoptosis in Huh-7-derived tumors was induced by the treatment with lenvatinib and losartan, and the combination of both agents profoundly enhanced the induction of cellular apoptosis (Figure 4A,C). Huh-7-derived xenograft tumors showed a higher expression of *AGT1R* than human liver tissues, indicating that the losartan-mediated inhibition of tumor growth ascribed to the blockade of AT-II and AT1R signaling (Figure 4D).

### 3.4. Combined Effects of Lenvatinib and Losartan on Intratumor Angiogenesis in Liver Cancer Cells

Given the in vitro suppressive effects of lenvatinib and losartan on angiogenic activity, we next investigated the combination effects of both agents on intratumor angiogenesis in Huh-7-derived xenograft tumors. Immunohistochemical analysis demonstrated that newly formed CD34-positive intratumor vessels were significantly decreased in mice receiving either single treatment with lenvatinib (3 or 10 mg/kg) or losartan (Figure 5A,B). The combination of both agents significantly augmented the reduction in CD34-positive vessels as compared to either single treatment (Figure 5A,B). A semiquantitative analysis showed that the combination of losartan with a lower dose of lenvatinib (3 mg/kg) could exert an anti-angiogenic effect to the same extent as that achieved with a higher dose of lenvatinib (10 mg/kg) (Figure 5A,B). In accordance with a decrease in CD34-positive vessels, *CD34* mRNA levels were lower in xenograft tumors of mice treated with lenvatinib (3 or 10 mg/kg) or losartan (Figure 5C). In accordance with the results that losartan could reduce the VEGF-A production in liver cancer cells, intratumor expressions of *VEGF-A* were decreased in losartan-treated mice (Figure 5D). Intratumor *VEGF-A* was overexpressed in lenvatinb-treated mice due to an autocrine VEGF and VEGFR positive-feedback signaling loop. Strikingly, combination with losartan efficiently attenuated the lenvatinib-induced autocrine *VEGF-A* overexpression (Figure 5D). Meanwhile, losartan did not change *FGF2* expression or affect lenvatinib-induced *FGF2* overexpression (Figure 5E). Moreover, we found that intratumor C-X-C motif ligand 8 (*CXCL8*) expression was reduced in losartan or lenvatinib-treated mice, but the combination of both agents did not show a more potent reduction (Figure 5F).

## 4. Discussion

In the present study, we provide the first evidence that losartan potently augmented the anticancer effects of lenvatinib against human liver cancer cell growth. Our results showed that losartan efficiently suppressed AT-II-stimulated cell proliferation and induced cell apoptosis in several human liver cancer cell lines. It is well documented that AT-II can promote tumor growth in HCC. Previous reports have elucidated several underlying mechanisms for the induction of HCC cell proliferation by AT-II. Ji et al. have shown that AT-II and AT1R signaling stimulates proliferation and inflammation of human HCC cells through activation of protein kinase C (PKC)/nuclear factor-kappa B (NF-κB) [27]. Qi et al. also have demonstrated that the AT1R/Raf/ERK1/2 pathway plays a key role in human HCC cell proliferation and identified Bcl-2 and c-Myc as downstream targets of this pathway [20]. Based on this mechanistic evidence, the pharmacological blockade of AT-II and AT1R has shown the anti-proliferative capacity on HCC cell growth. Losartan could reduce the diethylnitrosamine-induced HCC development in mice with cell cycle arrest through inactivation of the nuclear factor-kappa B (NF-κB) pathway [28]. Additionally, telmisartan, another type of ARB, was reported to induce G0/G1 cell cycle arrest and cause apoptosis in HCC cells [29]. Our in vitro study also indicated that the anti-proliferative effects of losartan on HCC cells were observed under AT-II stimulus but not without AT-II. Additionally, Huh-7-derived subcutaneous tumors showed a clear increase in AT1R protein level as compared to those of normal human liver tissues. As supported by these findings, the anti-proliferative properties of losartan arise from the inhibition of AT-II and AT1R signaling.

Since the AT-II/AT1R axis plays a key role in tumor angiogenesis, we focused on the impact of losartan on the changes in HCC angiogenic status and its interaction with lenvatinib-mediated angiogenic activity. Autocrine VEGF production is one of the key molecular factors to acquire resistance to VEGFR inhibitors in HCC cells [30]. A recent report has demonstrated that treatment with sorafenib increased VEGF production in human hepatoma cells [31]. In line with this, our results revealed the autocrine production of VEGF-A in Huh-7 cells under lenvatinib treatment. We also elucidated that the AT-II stimulus upregulated VEGF-A production in human liver cancer cells, which was reversed by losartan treatment. This finding coincides with previous evidence stating that candesartan downregulated VEGF expression by inhibiting the AT1R and VEGF pathway in human HCC cells [32]. Intriguingly, losartan also attenuated the lenvatinib-mediated VEGF-A production in three types of HCC cell lines, suggesting that combination with losartan may help prevent HCC cells from acquiring resistance to lenvatinib. Moreover, combination with losartan enhanced the inhibitory effects of lenvatinib on VEGF-mediated HUVEC proliferation and tubule formation. Altogether, our results demonstrate that AT-II and AT1R blockade could modulate anti-angiogenic properties of lenvatinib in both HCC cells and vascular endothelial cells.

In the clinical practice, as with other molecular targeted agents, a subset of patients may experience adverse effects from lenvatinib treatment [11,33,34,35]. Among various adverse effects mediated by lenvatinib, hypertension was one of the most common, and many patients often needed to be medicated with anti-hypertensive agents. Of note, our in vitro studies found that both anti-proliferative and anti-angiogenic effects of losartan combined with lenvatinib at a lower dose (1 μM) were approximately comparable to those of lenvatinib at a conventional dose (3 μM). Correspondingly, xenograft analyses also revealed the augmentation of lenvatinib-mediated antitumor effects by combination with losartan. Notably, losartan combined with lenvatinib (3 mg/kg) exerted tumor suppression comparable to that achieved with lenvatinib (10 mg/kg), in parallel with the results of in vitro studies. These results may suggest that combined lower doses of lenvatinib and losartan would confer the clinical benefits to patients who are intolerable to conventional doses of lenvatinib. In particular, other anti-hypertensive agents did not augment the lenvatinib-mediated anti-angiogenic effects in the in vitro HUVEC model. Thus, it is likely preferable to use ARBs in patients whose blood pressure is elevated due to administration of lenvatinib.

Though the findings detailed above do seem robust, several points require further clarification. First, we examined the combination effects of both agents on several human liver cancer lines, including Huh-7, HLE, and JHH-6. The losartan-mediated enhancement of antitumor properties would be dependent on AT1R expression levels in HCC cells. A recent study has demonstrated that AT1R was highly expressed in human HCC tissues as compared to those in normal adjacent tissue, and intratumor upregulation of AT1R is associated with HCC progression and pathological characteristics, including intrahepatic metastasis, portal vein invasion, TNM stage, and histological differentiation [36]. Therefore, further studies should address whether the effect of this combination therapy varies with AT1R expression levels. Second, an impressive report from Zhang et al. has claimed that ACE-Is did not lead to clinical benefits in proteinuria caused by anti-angiogenic drugs in HCC tumor-bearing mouse models [37]. Interestingly, they also showed that ACE-Is accelerated kidney-derived erythropoietin production, which could compromise the effects of anti-angiogenic drugs. To assess the conflicting effects of AT-II blockade on HCC, additional analyses including the measurement of erythropoietin production levels are further required.

Collectively, we demonstrate that additive treatment with ARB enhances the tumor suppressive effects of lenvatinib in the human cancer cell-derived xenograft model. We emphasize that the combined treatment could provide advantageous outcomes to patients with intolerance to a common dose of lenvatinib due to adverse effects including hypertension. Given that ARBs are clinically available without severe toxicities, they may eventually emerge as viable modulators of molecular targeted agents for patients with advanced HCC.

## Figures and Tables

**Figure 1 cells-10-00575-f001:**
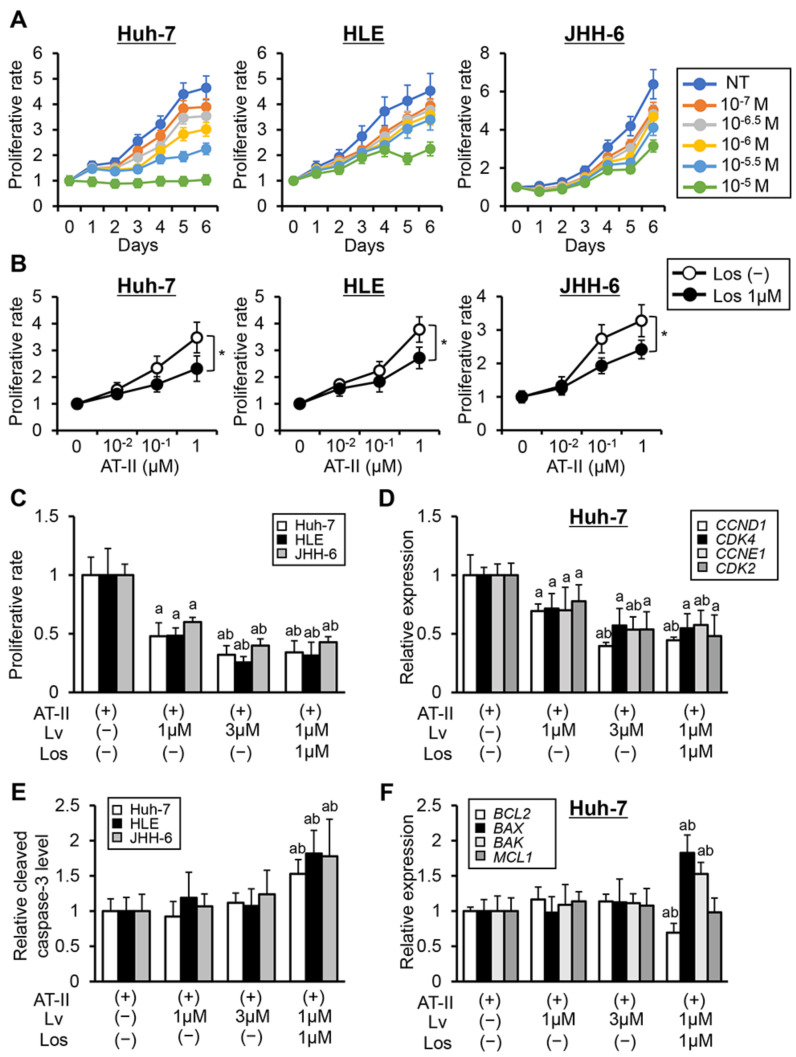
In vitro cytostatic effects of lenvatinib and losartan on liver cancer cells. (**A**) Cell proliferation of human liver cancer cells (Huh-7, HLE, and JHH-6) incubated with lenvatinib (0–10^-5^ M) for 0–6 days. (**B**) Cell proliferation of human liver cancer cells pre-treated with different concentrations of angiotensin-II (AT-II) (0–1 μM) for 12 h and subsequently treated with losartan (Los) (1 μM) for 12 h. (**C**–**F**) Cell proliferation of human liver cancer cells (**C**), relative mRNA expression levels of cell cycle-related markers in Huh-7 (**D**), the levels of cleaved caspase-3 in human liver cancer cells culture extract assessed by ELISA (**E**), relative mRNA expression levels of apoptosis-related markers in Huh-7 (**F**), cells were pre-treated with AT-II (1 μM) for 12 h and subsequently treated with Los (1 μM) and lenvatinib (Lv) (1 or 3 μM) for 12 h. The mRNA expression levels were measured by qRT-PCR, and *GAPDH* was used as internal control (**D** and **F**). Quantitative values are relatively indicated as fold changes to the values of (**A**) group at the start of treatment with lenvatinib in each dose, (**B**) group of AT-II (0 μM) in each dose of Los (**B**), (**C**–**F**) group of AT-II(+)/Lv(-)/Los(-). Data are mean ± SD (*n* = 3 independent experiments with *n* = 8 samples per condition). * *p* < 0.05 indicating a significant difference between groups (**B**). ^a^
*p* < 0.05, ^b^
*p* < 0.05 compared with group treated with AT-II(+)/Lv(-)/Los(-) and AT-II(+)/Lv(1 μM)/Los(-), respectively (**C**–**F**).

**Figure 2 cells-10-00575-f002:**
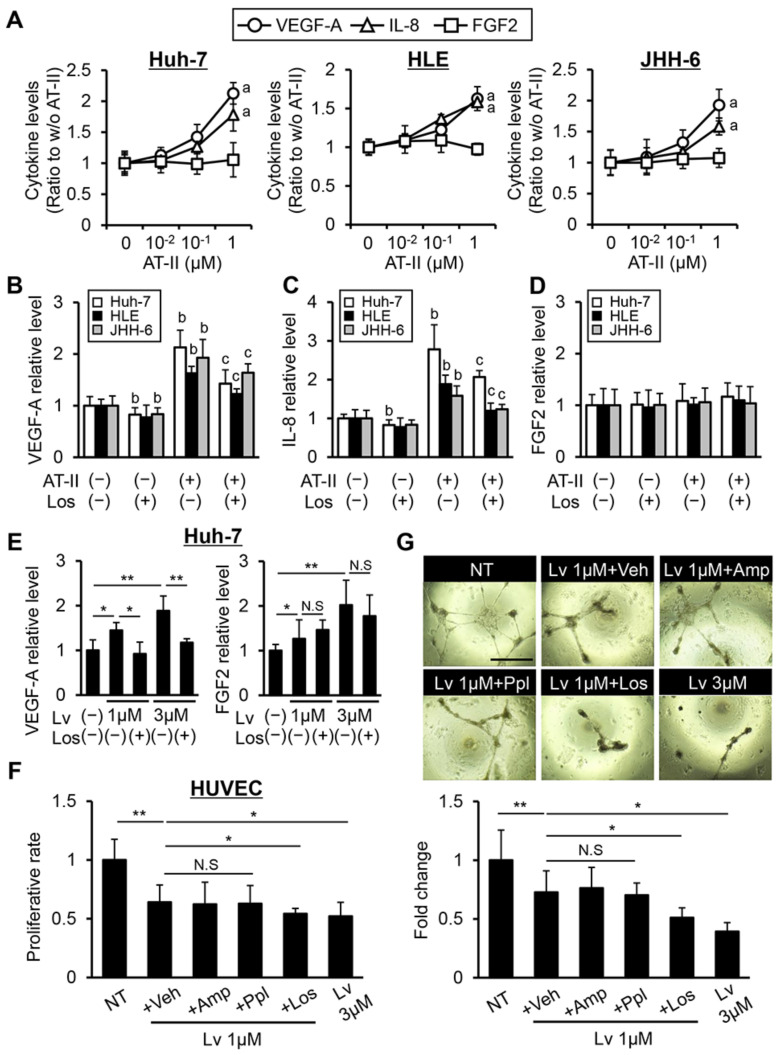
In vitro angiostatic effects of lenvatinib and losartan on liver cancer cells and endothelial cells. (**A**) VEGF-A, IL-8, and FGF2 levels in human liver cancer cells (Huh-7, HLE, and JHH-6)-cultured media assessed by ELISA. Cells were treated with different concentrations of angiotensin-II (AT-II) (10^−2^–1 μM) for 12 h. (**B**–**D**) VEGF-A (**B**), IL-8 (**C**), and FGF2 (**D**) levels in human liver cancer cells-cultured media. Cells were pre-treated with AT-II (1 μM) for 12 h and then treated with losartan (Los) (1 μM) for 12 h. (**E**) VEGF-A and FGF2 levels in Huh-7-cultured media. Cells were treated with Los (1 µM) and lenvatinib (Lv) (1 or 3 μM) for 24 h. Quantitative values are relatively indicated as fold changes to the values of group of Los(-)/Lv(-). (**F**) Cell proliferation of human umbilical vein endothelial cells (HUVEC) assessed by ELISA. HUVEC cells were stimulated with VEGF-A (10 ng/mL) and concomitantly treated with Lv (1 μM) and vehicle (Veh: DMSO)/ amlodipine (Amp: 10 μM)/ propranolol (Ppl: 50 μM)/ Los (1 μM) or with Lv (3 μM) alone for 24 h. (**G**) Characteristics (Upper panels) and index (Lower panel) of in vitro HUVECs tubular formation. HUVECs were cultured under the same conditions as (**F**). Scale bar; 100 μm. Micro-vessels index was quantified in high-power field by ImageJ software. Quantitative values are relatively indicated as fold changes to the values of (**A**) group of AT-II (0 μM), (**B**–**D**) group of AT-II(-)/Los(-), (**E**) group of Los(-)/Lv(-), (**F** and **G**) group of non-treatment (NT) (only with VEGF-A stimulation). Data are mean ± SD (*n* = 3 independent experiments with *n* = 8 samples per condition). ^a^
*p* < 0.05 compared with group treated with AT-II (0 μM) (A), ^b^
*p* < 0.05, ^c^
*p* < 0.05, compared with group treated with AT-II(-)/Los(-) and AT-II(+)/Los(-), respectively (B-D), * *p* < 0.05; ** *p* < 0.01 indicating a significant difference between groups (**E**–**G**). N.S: Not significant.

**Figure 3 cells-10-00575-f003:**
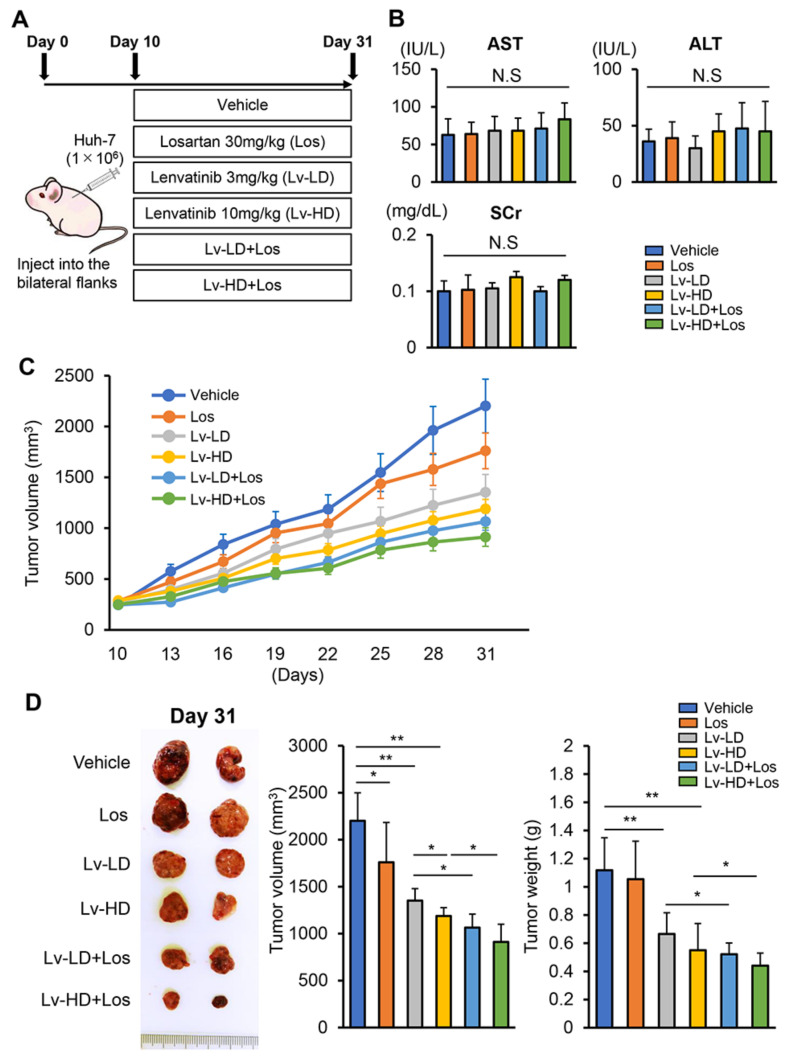
Antitumor effects of lenvatinib and losartan on HCC-derived xenograft tumor. (**A**) Experimental protocol. (**B**) Serum levels of aspartate aminotransferase (AST), alanine aminotransferase (ALT) and creatinine (SCr) in the experimental mice groups at the end of experimental period. (**C**) Time course of Huh-7-grafted subcutaneous tumor volumes in the experimental groups. (**D**) Representative photograph of resected subcutaneous tumors (Left panel), the mean tumor volumes (Middle panel) and weight (Right panel) in the experimental groups at the end of experimental period. Data are mean ± SD (*n* = 20 tumors/10 mice). **p* < 0.05; ***p* < 0.01 indicating a significant difference between groups. N.S: Not significant.

**Figure 4 cells-10-00575-f004:**
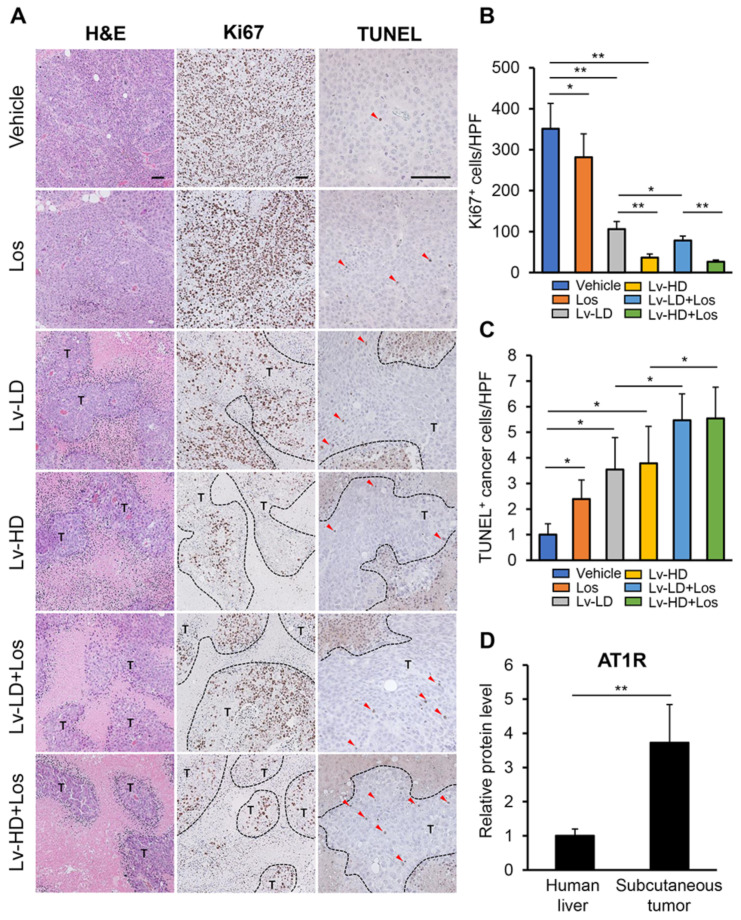
Intratumor cell proliferation and apoptosis in HCC-derived subcutaneous tumors. (**A**) Representative pictures of Huh-7-grafted subcutaneous tumors stained with H&E, Ki67, and TUNEL. T; tumor lesions, Red triangles indicate intratumor apoptotic cells. Scale bar; 100 μm. (**B**) Quantification of Ki67^+^ cells. The number of immunopositive cells in high-power field (HPF) were counted for quantification. (**C**) Quantification of TUNEL^+^ cells. The number of immunopositive cells in HPF were counted for quantification. (**D**) Protein levels of AT1R in human normal liver tissue and Huh-7-grafted subcutaneous tumors assessed by ELISA. Data are mean ± SD (*n* = 20 tumors/10 mice; **B** and **C**, *n* = 5; **D**). * *p* < 0.05; ** *p* < 0.01 indicating a significant difference between groups.

**Figure 5 cells-10-00575-f005:**
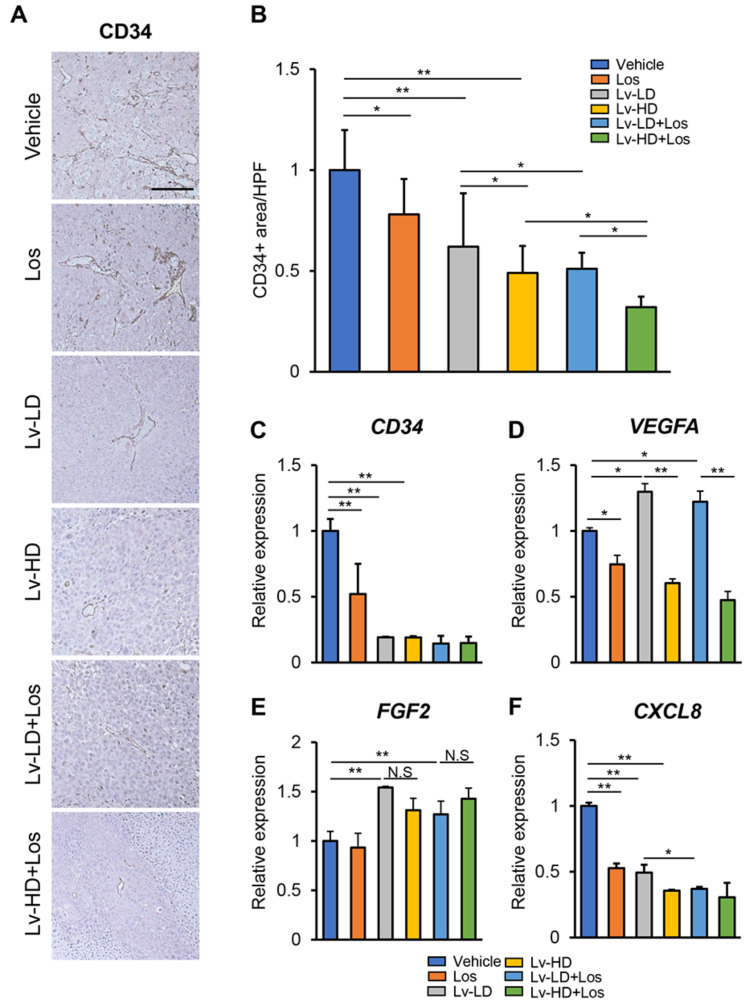
Intratumor angiogenesis in HCC-derived subcutaneous tumors. (**A**) Representative microphotographs of CD34 staining in the Huh-7-grafted subcutaneous tumors. Scale bar; 100 μm. (**B**) Semi-quantitation of CD34-positive vessels in the experimental groups in high-power field (HPF) by ImageJ software. Quantitative analysis included five fields per section and quantitative values are relatively indicated as fold changes to the values of vehicle group. (**C**–**F**) Relative mRNA expression of (**C**) *CD34*, (**D**) *VEGFA,* (**E**) *FGF2* and (**F**) *CXCL8* in the Huh-7-grafted subcutaneous tumors. The mRNA expression levels were measured by qRT-PCR, and *GAPDH* was used as internal control. Data are mean ± SD (*n* = 20 tumors/10 mice; **B** and **C**, *n* = 5; **D**). **p* < 0.05; ***p* < 0.01 indicating a significant difference between groups. N.S: Not significant.

## Data Availability

The data that support the findings of this study are available from the corresponding author upon reasonable request.

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
