# Peer review of "The Angiotensin II Receptor Blocker Losartan Sensitizes Human Liver Cancer Cells to Lenvatinib-Mediated Cytostatic and Angiostatic Effects"

_cells, 2021, doi:10.3390/cells10030575_

Round 1
Reviewer 1 Report
The article is good , the figures are clear. You worked very well.
Please you have to correct at 44 line control rate,
then I would like to suggest you that you have to explain and report how you linked cell culture concentration and dose oral intake of drugs cited in the paper and if you measured concentration of drugs i used animals.
Author Response
We appreciate the reviewer for pointing out the error in line 44. We corrected these parts. Also, we really thank the reviewer for suggesting the important points. In this study, we did not perform pharmacokinetics and pharmacodynamics, and we defined doses oral intake of losartan and lenvatinib in reference to previous evidences. As for the dose of losartan, we have investigated the antitumor effect of losartan at the same dose (30mg/kg for in vivo and 1μM for in vitro) on human cholangiocarcinoma xenograft model using same nude mice (Cancer Lett. 2018 Oct 10;434:120-129; New Ref18), and we confirmed that this dose did not show any toxicity. Regarding the dose of lenvatinib, we referred two previous reports which examined antitumor effects of lenvatinib on human HCC xenograft (Biochem Biophys Res Commun. 2019 May 21;513(1):1-7., Anticancer Res. 2019 Nov;39(11):5973-5982. New Ref 23 and 24). Both reports investigated its effects by using lenvatinib at 3, 10, 30 mg/kg as lower, common and higher dose, respectively. We added these references in Materials and Methods (line 147-149).
Reviewer 2 Report
The report by Hirotetsu Takagi et al, entitled: The angiotensin II receptor blocker losartan sensitizes human liver cancer cells to lenvatinib-mediated cytostatic and angiostatic effects, described interesting informations about human liver cancer cells. In my opinion the discussion section can be shortend.
Author Response
We appreciate for the reviewer for kind suggestion. According to the reviewer’s comments, we shortened the discussion section.